



# Ground mobile observation system for measuring multisurface microwave emissivity

Wenying He [1*,2]   Hongbin Chen[1,2]   Yuejian Xuan[1]   Jun Li[1]   Minzheng Duan[1,2]

1.Key Laboratory of Middle Atmosphere and Global Environment Observation, Institute of
Atmospheric Physics, Chinese Academy of Sciences, Beijing 100029, China
2.University of Chinese Academy of Sciences, Beijing 100049, China

**Abstract**

Large microwave surface emissivities with a highly heterogeneous distribution make it challenging to use satellite microwave data to retrieve precipitation and to be assimilated into numerical models over land. To better understand the microwave emissivity over land surfaces, we designed and established a ground observation system for the in situ observation of microwave emissivities over several typical surfaces. The major components of the system include a dual-frequency polarized ground microwave radiometer, a mobile observation platform, and auxiliary sensors to measure the surface temperature and soil temperature and moisture; moreover, observation fields are designed comprising five different land surfaces.

Based on the observed data from the mobile system, we preliminarily investigated the variations in the surface microwave emissivity over different land surfaces. The results show that the horizontally polarized emissivity is more sensitive to land surfaces than is the vertically polarized emissivity: the former decreases to 0.75 over cement and increases to 0.90 over sand and bare soil and up to 0.97 over grass. The corresponding emissivity polarization difference is obvious over water (>0.3) and cement (approximately 0.25) but reduces to 0.1 over sand and 0.05 over bare soil and almost 0.01 or close to zero over grass; this trend is similar to that of the Tb





polarization difference. At different elevation angles, the horizontally/vertically
polarized emissivities over land surfaces obviously increase/slightly decrease with
increasing elevation angle but exhibit the opposite trend over water.
Key words: Ground mobile observation system, microwave radiometer, microwave surface
emissivity, surface temperature, land surface

**1 Introduction**

The land surface microwave emissivity varies but is generally high ($\sim 0.90$) and

thus generates strong surface radiance; however, this strong surface radiance obscures
the atmospheric radiance, making it more difficult to assimilate and precisely retrieve
atmospheric parameters using satellite microwave data over land (McNally et al.,
2000; Farbou et al., 2005; Schwartz et al., 2012). Moreover, due to complex variations
affected by many surface factors, such as soil type, wetness, vegetation type and
surface roughness, the land surface emissivity is poorly understood. Hence, the land
surface microwave emissivity constitutes a major parameter limiting the application
of spaceborne microwave data over land.

Microwave emissivity models have been developed only for a limited range of

frequencies and surface conditions. For example, the emissivity over bare soil was
modeled at lower frequencies, and the soil dielectric constants were obtained from
ground-based measurements (Wang and Schmugge, 1980). Furthermore, the
emissivity over the vegetation canopy was simulated using a radiative transfer model
with a large number of canopy optical parameters (Mo and Schmugge, 1987; Isaacs et
al., 1989; Fung, 1994). Weng (2001) developed a microwave land emissivity model to



quantify the emissivity over various surface conditions, including snow, deserts, and
vegetation. Xie et al. (2017) developed a parameterized soil surface emissivity model
for bare soil surfaces and compared with Weng's model, results reflected the reduced
overall errors, especially for horizontal polarization. Ultimately, the microwave
emissivity of land surfaces is determined mainly by the soil dielectric constant, which
is influenced by the physical temperature, soil texture and moisture content, and
vegetation structure and type. As a result of these complicated parameters with
numerous uncertainties, establishing a common physical emissivity model and
accurately obtaining emissivity estimates by using only an emissivity model remain
challenging.

Satellite observations offering extensive coverage have been used to estimate the

regional and global distributions of land surface emissivity since the 1990s (Prigent et
al., 2000; Moncet et al., 2011). To avoid the impacts of the complex variability of
clouds and precipitation in the atmosphere, only the brightness temperatures observed
by spaceborne microwave instruments under clear sky conditions are generally
selected to calculate the land surface microwave emissivity. Jones and Vonder Haar
(1997) used SSM/I (Special Sensor Microwave Imager) microwave observations and
GOES/VISSR (Geostationary Operational Environmental Satellite/Visible Infrared
Spin-Scan Radiometer) infrared data that were closely matched in both space and time
to retrieve the microwave land emissivity over the Central United States and utilized
the infrared data with a constant infrared emissivity of 0.98 to calculate the land skin
temperature (LST) under clear sky conditions. Further, Ruston and Vonder Haar (2004)



directly employed spatially varying infrared surface emissivities in the retrieval of
LST to calculate the microwave emissivity and discovered that the
atmospheric-corrected microwave surface emissivity is valuable for determining land
surface characteristics but is sensitive to rain events. Prigent et al. (1997, 1999)
calculated the land surface microwave emissivity over Africa, some parts of Europe
and West Asia by combining SSM/I data with LST observations provided by ISCCP
(International Satellite Cloud Climatology Project). With subsequently improved
ISCCP LST and cloud product data, Prigent et al. (2006) presented a global land
surface microwave emissivity database retrieved from 10 years of SSM/I data and
plotted the monthly average land surface microwave emissivity onto a geographic
map. In their work, the microwave emissivity retrieval was based primarily on
radiative transfer calculations, in which infrared data were used to determine the LST
under clear sky conditions, and atmospheric sounding data were used to take the
effects of atmospheric attenuation into account. Nevertheless, due to the complexity
and variability of clouds and atmospheric precipitation, land surface microwave
emissivity estimates derived from satellite observations are available only under clear
sky conditions. Moreover, the cloud screening and LST retrieval methods still contain
numerous uncertainties, which represent the main sources of errors in emissivity
calculations.

At present, the accuracy of surface emissivity estimates calculated from either

emissivity models or satellite observations is limited by the complexity of the land
surface and the variability of vegetation types and soil moisture. Hence, surface



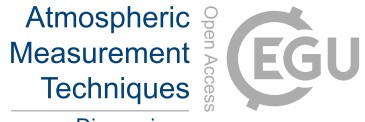

emissivity calculations need to be verified and improved with more in situ observation
data.

To better understand the variation characteristics of surface emissivity with

surface conditions, Ulaby et al. (1985) combined field experiments and theoretical
research and revealed that the land surface microwave specific emissivity is strongly
correlated with the distributions of soil moisture and vegetation. In addition, a few
observation experiments using ground-based microwave radiometers have been
carried out since the 1990s to study the variation characteristics of emissivity over
different surfaces (Njoku and O'Neill,1982; Matzler, 1990, 1994; Calvet, 1997;
Wigneron, 1994; Morland et al., 1995). More recently, in situ passive microwave
radiometer measurements over snow cover and sub-Arctic frozen soil have been used
to validate empirical emission models (Lemmetyinen et al., 2015; Montpetit et al.,
2018). Additionally, an aircraft-flown microwave radiometer was used to directly
observe the surface emissivity over forests, crops, snow and ice to analyze the
sensitivity of those emissivities to the view angle, frequency, measurement time and
surface characteristics (Hewison, 2001; Wigneron et al., 1997; Hewison and English,

1999).

The observation mode of a microwave radiometer in a field experiment is an

important consideration. Usually, ground-based radiometers are fixed when scanning
the observed field; for example, they can be mounted on a truck or a tower (Matzler,
1990; Lemmetyinen et al., 2015), allowing the instrument to better determine the
temporal evolution of surface emissivity over single type of land-cover area. In



contrast, using a mobile mode, such as airborne and mobile sled-based radiometers
(Morland, 2003; Lemmetyinen et al., 2015; Montpetit et al., 2018), can better reveal
the spatial evolution of surface emissivity over different land-cover areas, but it is not
easy to obtain long-term emissivity observations due to the high cost and effort.

To obtain the long-term temporal evolution of surface emissivity over different

types of surfaces simultaneously, we proposed and developed a ground mobile
observation system to enhance in situ microwave emissivity observations. Long-term
continuous emissivity field experiments can help to more accurately understand the
characteristics of passive microwave polarized emissivities over typical land surfaces,
form a benchmark for verifying the retrieved emissivities from satellite or emission
models, and establish an emissivity parameterization scheme for a given surface in
radiance assimilation. The outline of this paper is as follows: the design of the ground
mobile observation system for measuring surface emissivity is introduced in section 2;
the data and method used for the emissivity calculations are described in section 3;
then, the surface emissivity estimates obtained directly from the observation system
are discussed preliminarily in section 4; and a final short summary is given in section

5.

**2. Ground mobile observation system for surface microwave emissivity**

To obtain the surface emissivity over several typical surfaces simultaneously, we

designed a ground mobile observation system to carry out long-term field experiments
over 5 test plots. Fig. 1 is an on-site photo of the observation system operating at the
Xianghe observation site (116.98° E, 39.76° N), Hebei Province, China. As shown in





Fig. 1, the mobile observation system consists of five main parts: a dual-polarized
ground-based microwave radiometer to observe the surface and sky radiances, a
mobile platform to move back and forth along a track, and three auxiliary sensors to
measure the surface temperature, soil temperature and moisture. The observation field
includes five test plots, namely, water, cement, sand, bare soil and grass. From the
observation system, we can directly obtain surface microwave emissivity estimates
more accurately than is possible from satellite data or emissivity models, which is
important to properly understand the variation characteristics of land microwave
emissivities and to improve the emissivity parameterization schemes used in models.

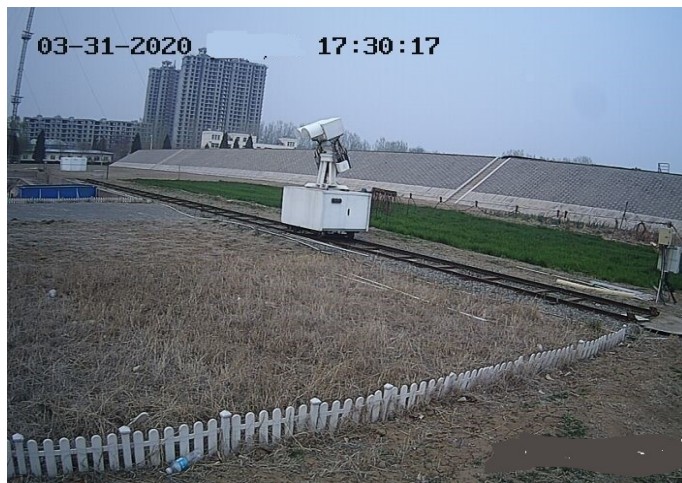


Fig 1 On-site photo of the surface microwave emissivity observation system
operating over various surfaces at the Xianghe site, China

**2.1 Ground-based microwave radiometer**
The core device of the observation system is a dual-frequency (18.7 and 36.5
GHz), dual-polarized (horizontal and vertical) microwave radiometer (RPG-4CH-DP)
produced by Radiometer Physics GmbH, Germany. The RPG-4CH-DP radiometer is a





high-performance instrument with a direct detection receiver and a completely
automatic calibration system. The radiometer is mounted on an accurate
elevation/azimuth positioner so that the whole system can perform scans in any
direction from the sky to the ground, thereby realizing complex scanning schemes,
such as all-sky monitoring and all-round monitoring of the ground. The RPG-4CH-DP
can distinguish cloud/raindrop particles during precipitation and monitor soil moisture
and vegetation parameters by using signals with different polarizations. Both
frequencies of 18.7 GHz and 36.5 GHz have been widely combined to detect snow
depth and snow water content and are frequently used in most spaceborne microwave
imagers, such as the SSM/I, AMSR-E (Advanced Microwave Scanning Radiometer
for EOS) and GMI (Greenhouse gases Monitoring Instrument) sensors. The directly
observed surface emissivities at these two frequencies can provide highly accurate
references for the verification and assimilation of spaceborne microwave
observations.
The RPG-4CH-DP radiometer has a comparable half-power beam width of
approximately 6° and a calibration accuracy of ±1 K. Currently, the height of the
instrument above the ground is 2.5 m, which results in a half-power footprint width of
0.22 m on average. More details regarding the instrument specifications for the
RPG-4CH-DP are shown in Table 1.


**Table 1 Instrument Specifications**

| Parameter | Specification |
|---|---|
| Radiometric resolution | 0.2 K RMS (1.0 s integration time) |
| Optical resolution | HPBW: 6.0° (Sidelobe level <-30 dBc) |



| | |
|---|---|
| Absolute system stability | 1.0 K |
| Receiver and antenna thermal stabilization | Accuracy <0.05 K |
| Pointing speed | Elevation: 3°/sec, azimuth: 5°/sec |
| Radiometric range | 0-350 K |
| Operating temperature range | -40°C to +45°C |
| Power consumption | <350 watts on average, 500-watt peak |
| Weight | 105 kg for receiver modules, 300 kg for positioner |


Currently, the RPG-4CH-DP provides only the basic brightness temperature (Tb)
data in 4 channels without other related products. By incorporating the auxiliary
observations from the observation system, we broadened the application of the
instrument, denoted RPG-XCH-DP, thereby providing not only the basic microwave
radiance but also the complex surface emissivity.
**2.2 Mobile system (platform)**
The multitarget mobile system comprises a track, a mobile platform, a driving
system and a control unit. As the sketch of the mobile system in Fig. 2 shows, the 25
m track is parallel to the test plots with an observation interval of 0.3 m. The mobile
platform placed on the track is a metal box 4 m in length, 0.8 m in height, and 1.0 m
in depth. The driving system includes a stepper motor, transmission mechanism, and
communication cable connected to the mobile platform and power supply. The control
unit consists of a single-chip microcomputer, timer and stepper motor driver, which
can set the moving time and control the operation of the driving device. The control
device is installed on the mobile platform and connects both driving devices.
In this experiment, to obtain the microwave emissivity over different surfaces in
near-simultaneous time, the RPG-4CH-DP is mounted on the mobile platform and
moves back and forth along the track. The communication system for receiving the
data and the power supply are placed in the metal box. According to the commands
from the single-chip microcomputer and the driving force from the stepper motor, the
mobile platform moves along the track similar to a small train, and the onboard



radiometer scans the 5 test plots at fixed times every day.

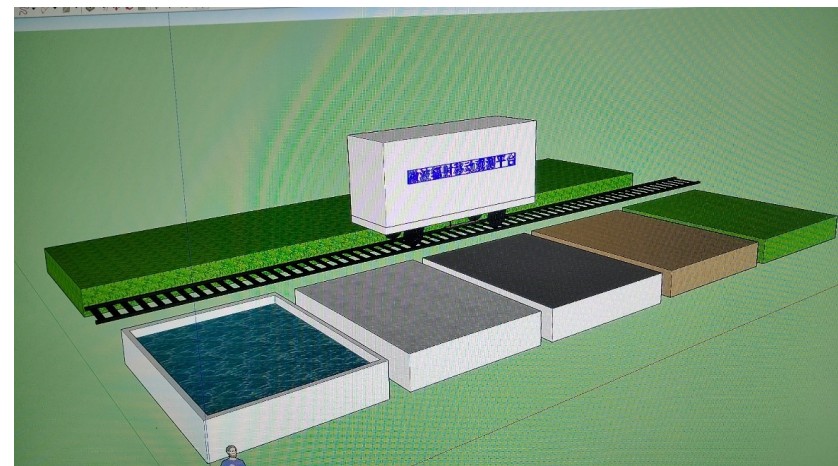


Fig. 2 Sketch of the mobile platform

**2.3 Observation field and auxiliary data**
Fig. 3 shows a sketch of the observation field, including the 5 test plots
distributed along the 25 m track. Currently, 5 surface types are considered in the
observation field, namely, water, cement, sand, soil and grass. For the water body, a
plastic pool 6 m long and 2.4 m wide is used to hold the water. The adjacent cement
surface consists of a 2 m wide footpath. The remaining three plots of sand, bare soil
and grass are the same size (approximately 6 m long by 4 m wide) and are separated
by a distance of approximately 2 m.

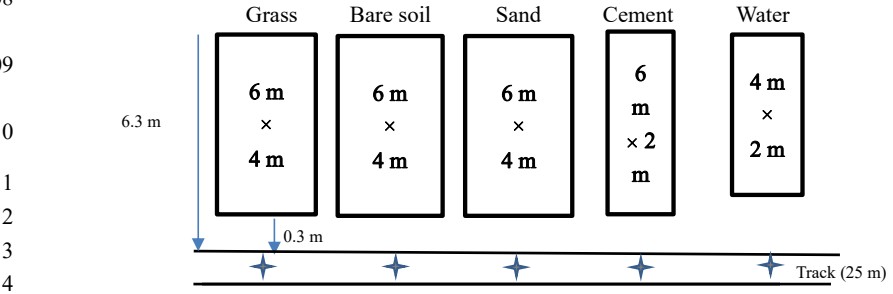

Fig. 3 Sketch of the observation field (including 5 test plots: water, cement, sand,
bare soil and grass), where   denotes the position of a touching switch

To scan each plot at the same place at a fixed time, five touching switches





corresponding to the center of each plot are fixed on the track to stop the moving
platform so that the radiometer can scan the same place for a couple of minutes. By
using this mobile platform, the ground-based radiometer can scan multiple surfaces
almost simultaneously (i.e., within 1 hr), thereby providing valuable measurements
for understanding the variation in surface emissivity over different land surfaces with
different characteristics.

The auxiliary data include mainly the surface temperature, soil temperature and

soil moisture. Five thermometers with a PT100 temperature sensor made by
Honeywell company are placed separately on each test plot to measure their surface
temperature. In addition, an SI-111 precision infrared radiometer developed by
Apogee Instruments Inc. is fixed on a stand of the RPG-4CH-DP radiometer to obtain
the surface temperature of each plot while the microwave radiometer is moving.
Furthermore, a set of soil temperature and humidity sensors is fixed at three soil
depths, 5 cm, 10 cm and 20 cm, to detect the subsurface soil temperature and humidity.
To monitor the real-time working situation of the whole observation system, a digital
video camera is installed near the field to record the states of the mobile platform and
radiometer as well as changes in the weather, such as the presence of cloud cover, rain
or snow.
**2.4 Scanning mode**

To directly obtain the surface emissivity, a combined mode of ground

observations at multiple elevation angles and zenith observations is designed, in
which the former monitors mainly the surface radiance while the latter monitors the
sky radiance in the same 1 hr period.

The ground observation mode is illustrated in Fig. 4. The mobile platform is

triggered every hour, and the microwave radiometer operates using the ground
scanning mode at this time. The scan is performed from the horizon (0°) to the ground,
and the elevation angle is defined as the angle between the scanning direction and the
horizontal. A negative value indicates an angle below the horizon, which is equivalent
to 90°-θ, where θ is the incident angle, an important parameter for describing
spaceborne radiometer scanning. The radiometer is 2.5 m above the ground, so it can





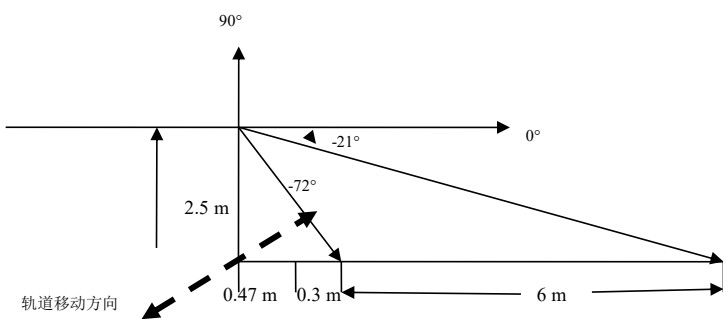

Fig. 4 Sketch of the combined scanning mode of the microwave radiometer

scan each test plot with a length of 6 m when the elevation angle is between -21° and
-72°, as shown in Fig. 4. The valid elevation angle range for water is different due to
the different length of the pool. To determine the surface emissivity variation with the
elevation angle, the radiometer is set to scan each test plot with an angle interval of 3°
from -21° to -45°, an angle interval of 5° from -45° to -70°, and then back to -21° to
scan the test plot repeatedly during the ground observation mode. To acquire ground
observations over all 5 test plots within 1 hr, each plot is given 9 minutes; in other
words, the mobile platform will move to the cement plot at 9 min, the sand plot at 18
min, the bare soil plot at 27 min, and finally the grass plot at 36 min. After finishing
the ground observations in all 5 test plots, the mobile platform will begin to move
back at 45 min and reach the beginning location after approximately 6 min. During
the return trip, the scan mode changes to the zenith observation mode so that the
radiometer scans from the ground to the sky. When the elevation angle is raised to 90°,
the radiometer will continually acquire zenith observations for approximately 5 min to
obtain the sky radiance. After obtaining these zenith observations, the elevation angle
changes from the zenith observation mode to the ground observation mode at -21° so
that the radiometer is already in the ground observation mode when the next
measurement cycle arrives. In this way, the radiometer on the mobile platform can
obtain not only the ground radiance over 5 test plots but also the sky radiance within a
1 hr period. Here, we assume that 1 hr is short enough to neglect the minute-scale



differences in the surface and sky radiance, and thus, the mobile system can obtain the
microwave emissivity over different surfaces nearly simultaneously.

**3 Data and method**
Three types of observation data are obtained from the field experiment: the
microwave brightness temperature (Tb) at different scanning angles from the ground
microwave radiometer; the surface temperature (Ts) of the five test plots measured
from the ground thermometers and infrared sensor; and the soil temperature and
moisture at three depths in the sand and bare soil plots.
When ground microwave radiometer scans the surface, the measured Tb comes
mainly from two contributions: that of upward radiation from the surface and that of
the reflected downward atmospheric radiance. Thus, the measured Tb can be
approximately expressed by Eq. (1):
$\qquad T_b = \varepsilon T_s + (1- \varepsilon) T_{sky}$ (1)
where $\varepsilon$ is the surface emissivity, $T_s$ is the surface temperature, and $T_{sky}$ is the radiance
from the sky. From Eq. (1), the surface emissivity can be directly calculated using Eq.
(2) by combining the Tb contributions from the surface and sky with the surface
temperature synchronously measured from the infrared sensor in the observation
system.
$\qquad \varepsilon=(T_b-T_{sky})/(T_s-T_{sky})$ (2)
Through applying the ground mobile observation system for surface microwave emissivity and
combining the video camera records with the soil temperature and moisture measurements, we can
not only directly obtain highly accurate surface microwave emissivity observations over different
test plots but also investigate the variation characteristics of the surface emissivity under different
weather conditions.

**4. Preliminary results**
Considering both the viewing field of the microwave radiometer and the size of the



test plots, the elevation angle range between -24° and -65° is chosen for observing the
land test plots (cement, sand, soil and grass), while elevation angles between -33° and
-65° are valid for observing the water surface. Here, we focus on the variations in the
radiance and surface emissivity over the 5 test plots during the observations recorded
in October 2018 under clear sky conditions.
**4.1 Radiance**
Since a scanning angle of 36° is equivalent to an incident angle of 54° used for
many spaceborne microwave imagers, such as AMSR-E (55°) or SSM/I (53°), we first
compare the variation in the observed Tb over different surfaces at an elevation angle
of 36°. The changes in the observed Tb at 36.5 GHz in horizontal (Tb36h) and vertical
(Tb36v) polarization over the four land surfaces within 24 hr (Beijing Time, BJT) are
quite similar, with smaller values at night and larger values at noon. Less variation in
the radiance is noted at Tb36v (not shown), but more significant variations are
detected at Tb36h over the four surfaces as shown in Fig. 5a: the observed Tb36h
from grass is approximately 270-285 K but varies within 240-270 K over sand and
bare soil and reaches only 200-230 K for cement. The observed Tb at 18.75 GHz
within 24 h shows similar variations with only slight changes among the different
land surfaces. Likewise, the corresponding polarization differences (V-H) of Tb
within 24 hr are very similar to one another, so both DTb18vh and DTb36vh at 02:00
(BJT) are shown in Fig. 5b, revealing a slight difference (close to zero) for grass but
considerably larger differences for water and cement (almost up to 70 K for water)
and smaller differences over sand and soil (below 30 K). In addition, the values of





DTb18vh are larger than those of DTb36vh. The Tb polarization difference is more
significant over water than over land and is closely related to the roughness of the
land surface. In addition, the roughness of grass is obviously larger than that of the
other three land surfaces and thus scatters more surface radiance and reduces the
polarization difference. Therefore, the observed Tb polarization differences over the
different surfaces shown in Fig. 5b appear reasonable, and the given quantitative
polarization differences for certain surfaces can serve as a valid reference for
identifying land surfaces and water bodies.

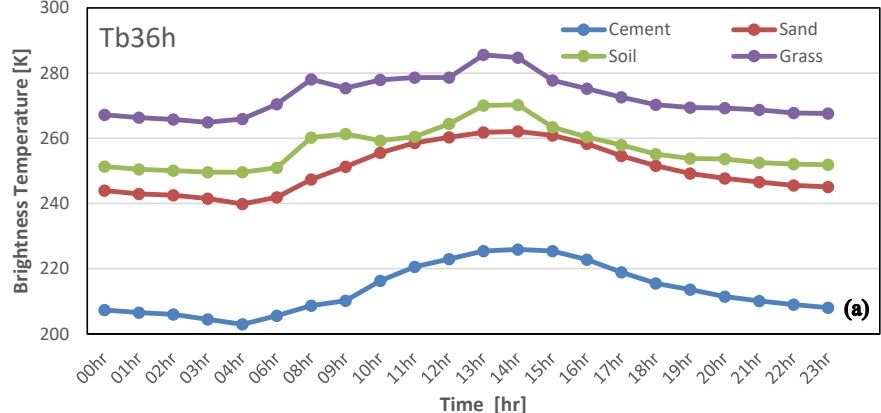


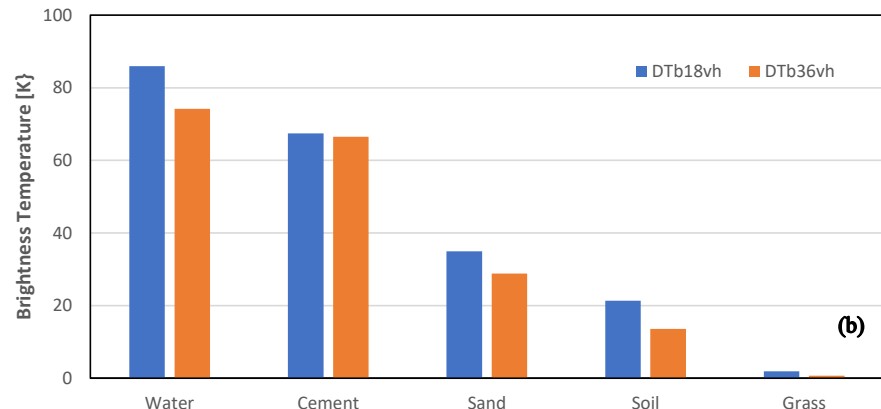




Fig. 5 Variations in the observed Tb (a) and Tb polarization differences (b) over
different surfaces in October 2018.
To study the variations in Tb at more than a single angle, Fig. 6a shows the changes in
the observed Tb with the elevation angle ranging from 24° to 65° over the four land
surfaces. The horizontally polarized Tb is clearly more sensitive to land surfaces than
the vertically polarized Tb with increasing elevation angle; in particular, Tb36h rises
rapidly from 180 K to 240 K over cement but slowly increases from 240 K to 260 K
over sand and bare soil and remains almost constant over grass. In contrast, the
variations in the vertically polarized Tb with increasing elevation angle are similar
among the land surfaces and are smaller than those in the horizontally polarized Tb,
showing a decreasing trend from 280 K to 260 K over different surfaces. In addition,
the variations in the observed Tb over water are presented in Fig. 6b. Different from
the above observations over land surfaces, the vertically polarized Tb over water
obviously reduces from 200 K to 140 K with increasing elevation angle, while the
horizontally polarized Tb slowly rises from 100 K to 120 K, almost opposite to the Tb
polarization variations over land surfaces. The corresponding changes in the
polarization difference of Tb at 18.75 GHz (DTb18vh) over all 5 classes of surfaces
are further plotted in Fig. 6c. In general, the Tb polarization difference decreases with
increasing elevation angle, and the variated ranges with the elevation angle over the 5
classes surfaces in Fig. 6c are similar to those in Fig.5b; thus, the decreasing trend is
most obvious over water and cement and lest evident over grass with increasing
elevation angle. Furthermore, the variations of the Tb polarization difference at 36.5





GHz with the elevation angle are similar to those at 18.75 GHz over all 5 test plots.

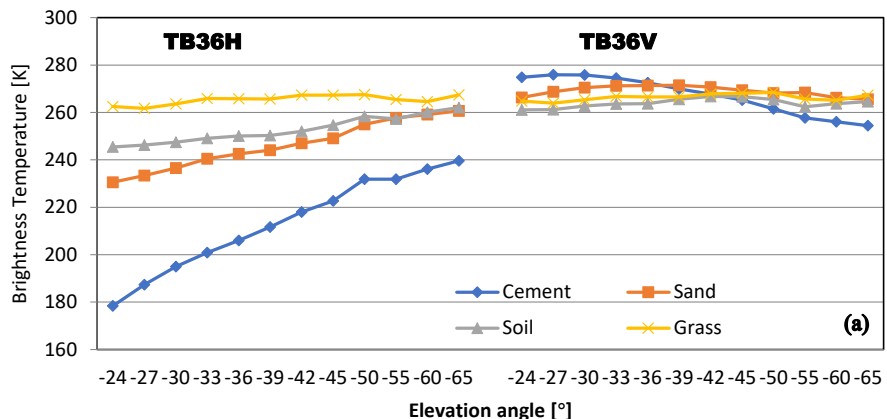


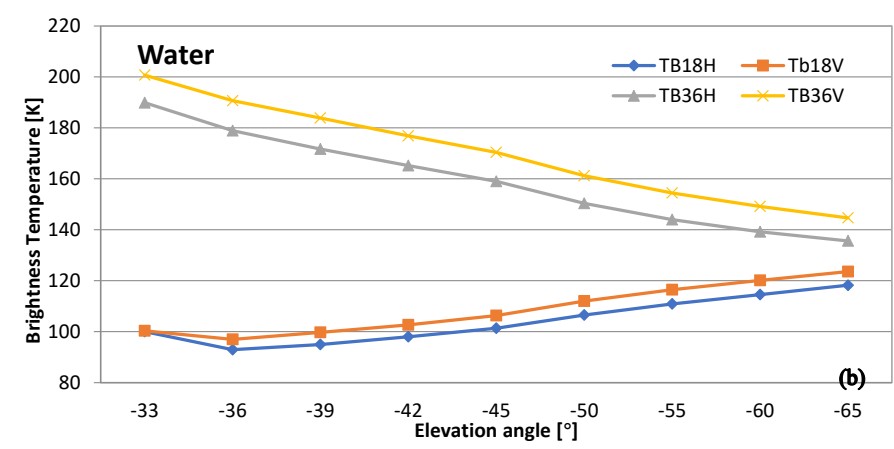


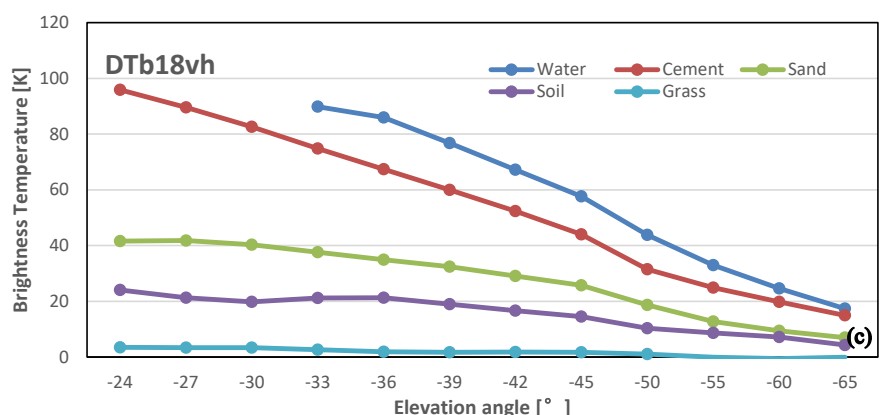


Fig. 6 Variations in the observed radiance over different land surfaces with the



elevation angle
**4.2 Surface microwave emissivity**
Combining the surface and sky radiance contributions to Tb with the surface
temperature derived from the infrared sensor, the surface emissivity ($\varepsilon$) is derived
from Eq. (2). Since the diurnal variation of $\varepsilon$ is more constant and less significant than
that of the Tb radiance, the surface emissivity observed at 02:00 (BJT) is chosen for
the following investigation. First, the polarized $\varepsilon$ at both 18.75 and 36.5 GHz and their
polarization differences at an elevation angle of 36° are compared in Fig. 7a. The
vertically polarized $\varepsilon$ ($\varepsilon_v$) is clearly much larger than the horizontally polarized $\varepsilon$ ($\varepsilon_h$),
and the $\varepsilon$ values at the same frequencies are close, but the $\varepsilon$ values over water is
smaller than those over the four land surfaces. The $\varepsilon_h$ values obviously differ among
the 4 land surfaces, although their corresponding $\varepsilon_v$ values are relatively similar,
exceeding 0.95, which indicates that $\varepsilon_h$ is more sensitive to land surfaces than $\varepsilon_v$. The
$\varepsilon_h$ is lower than 0.75 over cement but increases to 0.90 over sand and bare soil and up
to 0.97 over grass. Thus, the polarization difference ($\varepsilon_v$-$\varepsilon_h$) shown in Fig. 7b is
obvious over water (>0.3) and cement (approximately 0.25) but reduces to 0.1 over
sand and 0.05 over bare soil and almost 0.01 or close to zero over grass; this trend is
similar to that of the Tb polarization difference shown in Fig. 5b.

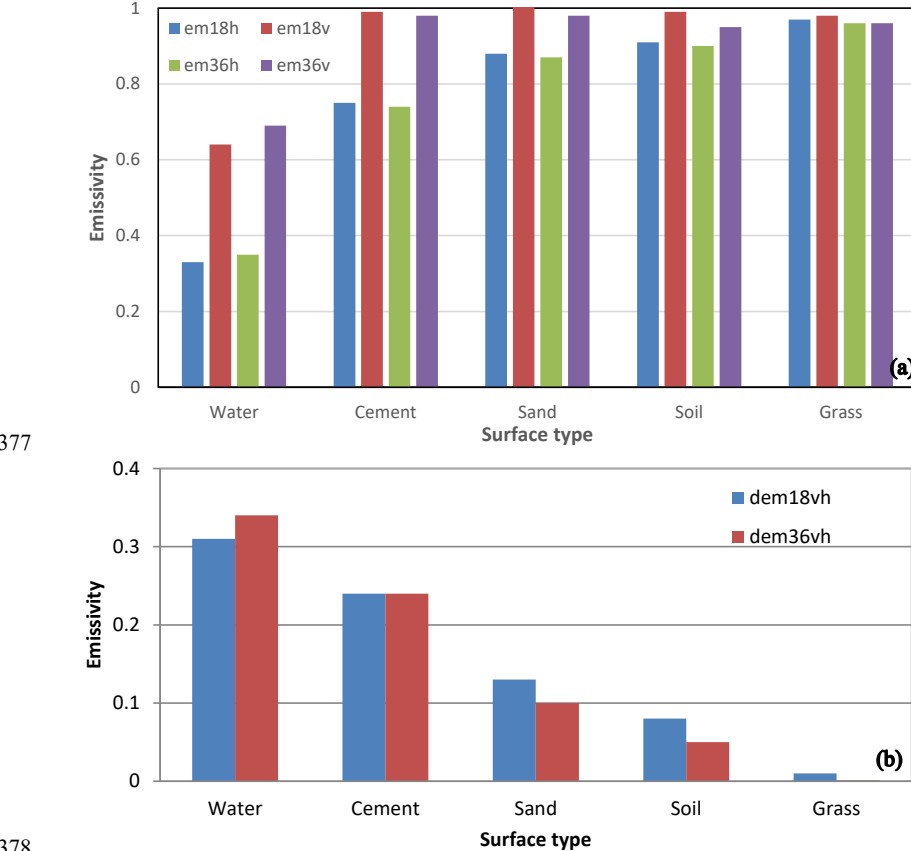



379  Fig. 7 Variations in the surface emissivity over different land surfaces at the

380 elevation angle of 36°

381  In addition to investigating the variations at a fixed angle, the variations in $\varepsilon$ at

382 multiple elevation angles over the 4 land surfaces are compared in Fig. 8a. Because $\varepsilon_h$

383 is more sensitive to land surfaces than to water, when the elevation angle changes

384 from -24° to 65°, $\varepsilon_h$ clearly rises from 0.65 to 0.85 over cement, followed by sand and

385 bare soil with $\varepsilon_h$ increasing from 0.85 to 0.95, and $\varepsilon_h$ is constant at 0.95 over grass.

386 The corresponding $\varepsilon_v$ values over the four land surfaces are closer and exhibit a

387 slightly decreasing trend within the range of 0.9-1.0 with increasing elevation angle.

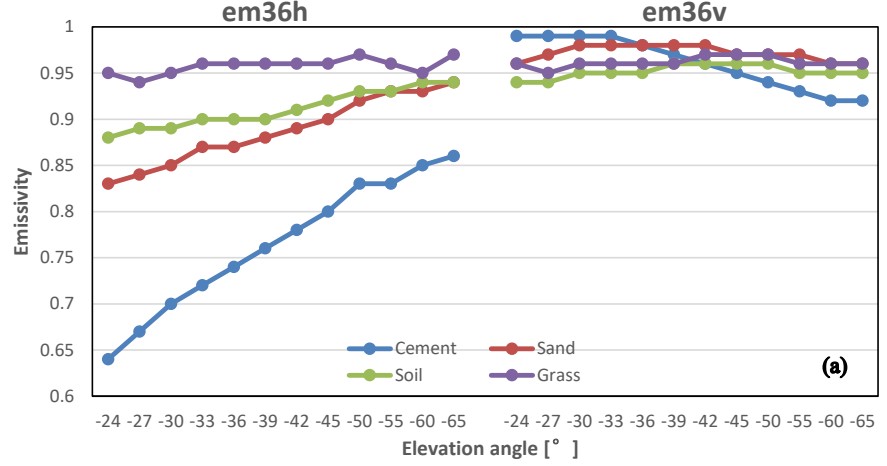


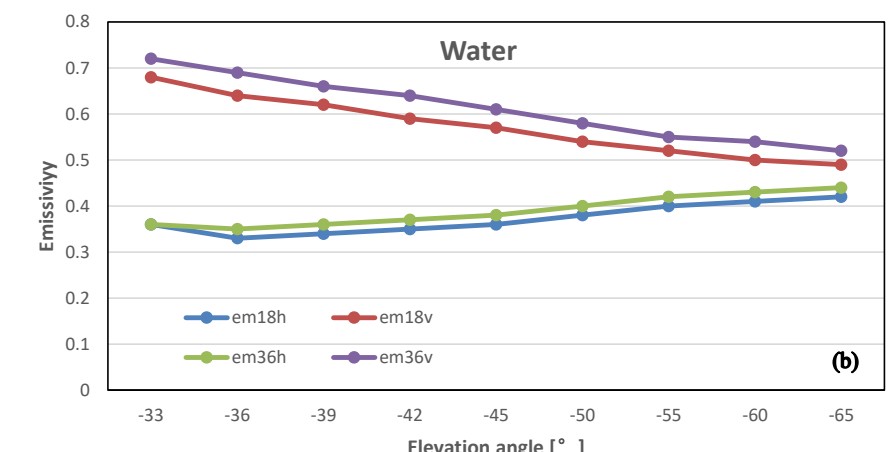


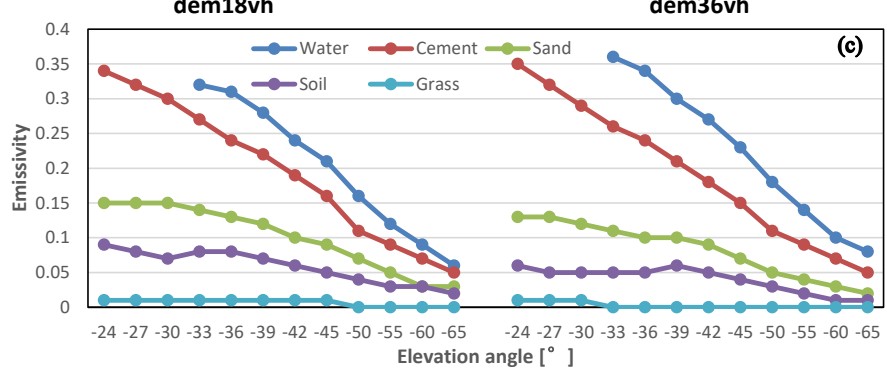


Fig. 8 Variations in the surface emissivity over different land surfaces(a,b) and

their ε polarization differences(c) with increasing elevation angle.





As shown in Fig. 8b, the ε values over water show considerably different variation
trends with the elevation angle from those over land surfaces: when the elevation
angle changes from -33° to -65°, $\varepsilon_v$ reduces from 0.7 to 0.5, while $\varepsilon_h$ slightly increases
within the vicinity of 0.4. The ε polarization differences ($\varepsilon_v$-$\varepsilon_h$) in Fig. 8c present
similar variation trends to those shown in Fig. 7b; that is, the ε polarization difference
decreases with increasing elevation angle, and the larger the polarization difference of
ε is (Fig. 7b), the greater the variation with the elevation angle (Fig. 8c). Hence, the
decreasing trend of ε is most obvious over water and cement, and ε slightly changes
over grass with increasing elevation angle. The variation in the ε polarization
difference at 36.5 GHz with the elevation angle is similar to that at 18.75 GHz over all
5 test plots (results not shown).

**405 5 Summary**

In this paper, we introduce a ground mobile observation system for directly
obtaining surface microwave emissivity estimates over five types of surfaces: water,
cement, sand, soil and grass. The mobile observation system consists mainly of a
dual-polarized ground-based microwave radiometer, a mobile platform, and auxiliary
sensors, and the observation field comprises 5 test plots.
Based on the observed data from the mobile system, we preliminarily
investigated the variation characteristics of the surface microwave emissivity over the
five different land surfaces. The results show that the horizontally polarized
emissivity is more sensitive to land surfaces than is the vertically horizontally polarized emissivity:



the former decreases to 0.75 over cement and increases to 0.90 over sand and bare soil
and up to 0.97 over grass. Hence, the corresponding polarization difference is obvious
over water (>0.3) and cement (approximately 0.25) but reduces to 0.1 over sand and
0.05 over bare soil and almost 0.01 or close to zero over grass; this trend is similar to
that of the Tb polarization difference. For different elevation angles, the
horizontally/vertically polarized emissivities over the land surfaces obviously
increase/slightly decrease with increasing elevation angle but exhibit the opposite
trend over water. Moreover, the emissivity polarization difference decreases with
increasing elevation angle, and the larger the emissivity polarization difference is over
a certain surface, the greater the variation with the elevation angle.
We developed a ground mobile observation system for measuring the microwave
emissivity over multiple surfaces, and the system has worked stably since August
2018. The preliminary results from our observation system partly reflect similar
variation trends to those reported by previous surface emissivity experiments, and
some are more related to the variation in emissivity at different elevation angles. In
future research, we will carry out further analyses and refine the emissivity
parameterization scheme for given surfaces based on long-term observations.
**Acknowledgments**
This work was supported by National Natural Science Foundation of China [No.
41575033] and National Key Research and Development Program of China
[2017YFC1501700]. We thank the staff at the Xianghe site for their maintenance
work on the microwave radiometer and the ground mobile observation system.

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
