# Peer review of "Ground mobile observation system for measuring"

_Atmospheric Measurement Techniques, 2021_

## Author Comment (AC1)

- **RC1**: 'Comment on amt-2021-165', Yanqiu Zhu, 25 Jul 2021  reply

This study carefully designed a ground observation system to investigate the variations of microwave emissivities over water and several typical land surfaces including cement, sand, bare soil, grass. This system's design made it feasible to avoid/minimize the uncertainties in the emissivity derivation caused by LST and atmospheric effect calculation and to assess the variations of emissivities over different surface types side-by-side in a controlled experiment environment. The topic of this manuscript is important to many applications of surface-sensitive radiances.

Specific comments:

1. The advantage of the ground mobile observation system is that it can provide temporal evolution of emissivity over different surface types at low costs in helping us to understand the characteristics of microwave emissivity, but in practice the usefulness of all these observations over different surface types may depend on the actual complexity of land surface in the area of observation site.

**Reply**:Thank you for your comments. Yes, the temporal evolution of emissivity over different surface types can be directly obtained from the ground mobile observation system at low costs, and the 5 test plots are designed to stand for certain typical surfaces. For the actual and complex land surface, it still need more real observations to find out.

2. Cloud and precipitation screening. The authors didn't provide any information on how they performed cloud and precipitation screening. Was this performed automatically on RPG-XCH-DP, or the authors used the video camera records or Tb information?

**Reply**: Yes, the video camera can record real-time weather conditions, such as rain or snow, and there is rain sensor on RPG-XCH-DP to detect rain or no-rain observation. In this paper we mainly present the observations in October 2018 under clear sky condition.

3. The major components of the system include a dual-frequency polarized ground microwave radiometer, a mobile observation platform, and auxiliary sensors to measure the surface temperature and soil temperature and moisture. The authors utilized the

observations from the ground microwave radiometer and surface temperature to derive emissivity. I notice that authors haven't used soil temperature and moisture observations. As the emissivity is determined mainly by soil dielectric constant, do the authors have any plan to use these soil observations, such as to study the relationship between emissivity and soil moisture?

**Reply**: Thank you for your concern. Yes, adding soil moisture observations is used to further study the relationship between emissivity and soil moisture. In this paper we focus on introducing the mobile observation system for surface emissivity and some preliminary results.

4. Brightness temperature Tb was referred to in many places in the manuscript. I assume authors meant the brightness temperature Tb in the ground observation mode.

**Reply**: Yes, in this paper Tb is the brightness temperature Tb in the ground observation mode.

5. Chinese characters appeared in Fig. 4. Please translate those into English.

**Reply**: Thanks for your care, it was translated into English in Fig.4.

**Citation**: https://doi.org/10.5194/amt-2021-165-RC1

---

## Author Comment (AC2)

- **RC2**: ['Comment on amt-2021-165'](), Anonymous Referee #2, 28 Jul 2021  reply

This work presents a well-designed emissivity observation experiment, including a dual-frequency, polarized microwave radiometer on a mobile platform as well as *in situ* measurements of coincident environmental parameters. Five different surface types are created and observed over a one month time period. The work is well performed and described. Some questions and comments amounting to minor revisions are given below.

-In the "real world" of satellite footprints, a homogeneous surface is rare. What are the authors' thoughts about what this experiment can tell us about emissivity variability in a heterogeneous field of view? Are there any plans to create something like this?

**Reply**: Yes, the "real world" of satellite footprint usually is a heterogeneous field, including many types of surface, such as soil, sand, and grass. Using this experiment can help us better know the temporal evolution of emissivity over certain typical surface. To obtain the emissivity over a heterogenous field of view, microwave radiometer can be installed at higher position, or mounted on a moving crane to observe the real and heterogenous field on the ground. We will try the plan later.

-Is this an ongoing experiment? It would be interesting to see data over a longer time period. Similarly, vegetation emissivty (higher than grass) is a key missing component here and vegetation life cycle would be a really interesting case to explore with this setup (though not necessary for this paper - something for future study).

**Reply**: Yes, it is ongoing experiment. It has been stably running since Sep. 2018, and we had obtained almost 2 years observations until now. In this paper we focus on introduce the mobile observation system, and will do more and detailed analysis using longer time observation in the later.

Thank you for your suggestion on the vegetation emissivity (higher than grass), such as wheat or other crops, we are more interested in that too. We did a couple of months observations for the wheat field on the other side of moving track in the last year, watching the grow of wheat from begin to end. We will do more data analysis for

those observations, and will plan to grow different crop in next year to see the different of vegetation emissivity. Thanks for your interesting, we will work on that in the future.

Abstract line 11: I realize this is the abstract, but a couple of words at the end of the first sentence as to why would be helpful here (e.g. due to the relatively small hydrometeor signal as compared to the land surface emission)

**Reply**: Thanks for your supplement, I did the change in the first sentence of abstract (line 9-10) as following "**Large microwave surface emissivities with a highly heterogeneous distribution and** the relatively small hydrometeor signal **over land make it challenging to use satellite microwave data to retrieve precipitation and to be assimilated into numerical models**."

Line 20-21: this occurs frequently in the paper that "sensitve to land surfaces" is used. I suggest changing these instances to "sensitive to surface type" or "sensitive to land surface variability" or similar.

**Reply:** Thank you for the comments. I did the modifications using the words you provided, such as line 21-22, line 337-338, and so on.

Line 35: "obscures radiance from the atmosphere and hydrometeors"

**Reply**: Thank you, it was modified into "**this strong surface radiance obscures radiance from the atmosphere and hydrometeors**" in line 35.

Line 45: Remove "Furthermore"

**Reply**: Yes, it was removed.

Lines 90-92: Another important limitation is availibility and accuracy of necessary input parameters on a global scale.

**Reply**: Yes, it was added in lines 92-93 as following "**At present, the accuracy of surface emissivity estimates calculated from either emissivity models or satellite observations is limited by the complexity of the land surface and the variability of**

vegetation types and soil moisture. Another important limitation is availability and accuracy of necessary input parameters on a global scale. Hence, surface emissivity calculations need to be verified and improved with more in situ observation data."

Lines 137-138: Add frequencies to this sentence.

**Reply:** It was changed in lines 138-139 into "a dual-frequency (18.7 and 36.5 GHz),dual-polarized ground-based microwave radiometer"

Line 164: Could also add another GMI here - the NASA GPM Microwave Imager also has these frequencies.

**Reply**: Thank you. It was modified as "such as the SSM/I, AMSR-E (Advanced Microwave Scanning Radiometer for EOS) and GMI (GPM Microwave Imager) sensors." In line 165-166.

Line 197: How many fixed times per day?

**Reply**: The fixed times are introduced more detail in "2.4 Scanning mode". Using 45min to scan 5 test plots in each hour, that is the 0thmin,$9^{th}$ min, $18^{th}$ min, $27^{th}$ min, and $36^{th}$ min of each hour are the fixed times for each test plot. If so 5 fixed times per hour, and 24*5=120 fixed times per day within 24hr.

Lines 200-207: This might be a good place to say something about penetration depth at these frequencies for each surface type.

**Reply**: Thanks for your suggestions. Penetration depth is a complex variable, hope to get more good ideas for that in the later.

Line 232: replace humidity with moisture

**Reply**: It was replaced.

Figure 4: Some Chinese characters on lower left

**Reply**: Thank you. It was modified in Fig.4.

Line 310: refer to Figure 5 here.

**Reply**: It was modified in Line 310 as "As Fig.5a shown, the changes in the observed Tb at 36.5 GHz".

Line 336: "more sensitive to the land surface type"

**Reply**: We did the change.

Line 342: Remove "In addition"

**Reply**: We removed it.

Line 353: Remove "Furthermore"

**Reply**: We removed it.

Figure 6: It might be interesting to add a line on these plots identifying the 36 degree (53-degree incidence) angle for reference

**Reply**: A good suggestion, we added a dotted line to identify the 36°in Fig.6.

Figure 7: Expand the caption with more information and label panels a) and b). Add information about time period of averaging and identify the b) panel as polarization difference

**Reply**: Thank you for your comment. We did the modifications in Fig.7 as "Fig. 7 Variations in the surface emissivity(a) and emissivity polarization differences (v-h) (b) over different land surfaces at 02:00 (BJT) in Oct. 2018"

Line 416: Remove "Hence"

**Reply**: It was removed.

Line 419: "the observed polarization difference"

**Reply**: It was modified.

Line 422: Remove "Moreover"

**Reply**: It was modified.

Line 403: Please include some discussion of differences between the Tb and emissivity plots, and why they occur

**Reply**: Thank you for your comment. We added some discussions in line 388-395 as following "Emissivity polarization differences is more significant over water than over land due to different surface reflectivity and dielectric constant property. Among four land surfaces $\varepsilon_v$-$\varepsilon_h$ over cement is most obvious and over grass is slight, which is closely related to land surface roughness. Both Tb and emissivity polarized difference demonstrated that surface roughness over grass is obviously larger than that over other three land surfaces, especially smooth cement surface, thus scatters more surface radiance and weakens the polarization difference over grass."

Line 414: "more sensitive to land surface type"

**Reply**: It was modified.

Variability with weather conditions is never really discussed - one would expect an emissivity decrease after precipitation due to water on the surface for example. Was this observed?

**Reply:** Thanks for your suggestion. Yes, we did observe the emissivity variability with weather conditions, such as obvious emissivity decrease in rainy case. In this paper we mainly introduce the mobile observation system and only show some results under clear-sky. We will do more comparisons for emissivity variability with weather conditions later.

**Citation**: https://doi.org/10.5194/amt-2021-165-RC2

---

## Author Comment (AC3)

- **RC3**: , Anonymous Referee #3, 01 Aug 2021  reply

The authors established a ground observation system to estimate the surface emissivities from brightness temperatures at four channels (18h, 18v, 36h, and 36v). The results are interesting. But the paper wasn't written clearly and the paper lack of some necessary information. The errors in the title of y-axis for figures 5,6,7,8 needs to be corrected.   The authors provide the measurement accuracy of the brightness temperature (1K), which is the specification of the radiometer. But, the authors didn't give the accuracy of the derived surface emissivities.

**Reply**: Thank you for your comments. We will try our best to modify the paper clearly and preciously.

The errors in the title of figures 5,6,7,8 have been corrected, sorry for our mistakes.

As to the accuracy of the derived surface emissivity, firstly it is obtained from the direct grounded observations, which help to reduce more uncertainties for estimated emissivity using model or satellite remote sensing data. Secondly, the microwave radiometer is the core device to obtain surface emissivity, and the measurement accuracy of observed brightness temperature from radiometer is basic and the most important guarantee for calculating surface emissivity. Third, to demonstrate the accuracy of derived surface emissivity better, we plan to do more comparisons between our results and emissivity model to check the consistency in theory although there are uncertainties in model.

Specific comments:

1. "Xie et al. (2017) developed a parameterized soil surface emissivity model for bare soil surfaces and compared with Weng's model, results reflected the reduced overall errors, especially for horizontal polarization." is unclear, whether Xie's model is more accurate?

**Reply**: Here we cited Xie et al. (2017) related work to show the development of emissivity model, and they mentioned that the improving results from the parameterized emissivity model were obtained on limited ground-based measurements and satellite data, and still need more validations and evaluations over larger areas and various surface conditions.

2. Define the emissivity polarization difference (vertically polarized – horizontally polarized?)

   **Reply**:Yes, it was added in line 377 as "the emissivity **polarization difference** ($\varepsilon_v$-$\varepsilon_h$) shown in Fig. 7b"

3. Change "angle" to "angles" in line 28.

   **Reply**: We did the change in line 28.

4. Explain why "but exhibit the opposite trend over water" in line 28.

   **Reply**: This sentence in line 28 is mainly coming from Fig.6(a,b). The different variations of surface emissivity over water and land with scan angle depend on the properties of surface reflectivity and **dielectric constant, and the detailed and quantitative emissivity values over different surfaces are obtained from direct** observation in this paper.

5. The soil emissivity depends on soil moisture and temperature. The authors mentioned the measurements in lines 231 and 232. The authors may provide the information in a table.

   **Reply**: Thanks for your concern. Yes, adding soil moisture observations is used to further study the relationship between emissivity and soil moisture. In this paper we focus on introducing the mobile observation system for surface emissivity, **and we will do more analysis for soil moisture data in the later.**

6. Water surface emissivity is a function of a surface wind. The surface wind is missed from the paper.

   **Reply**: Thank you for your suggestion. There is an automatic weather station nearby the observation site, wind data will be used to study the influence on water surface emissivity in the later.

7. (1) and (2) are good for a specular reflection. The authors may add sentences about why the surface reflection here is neither Lambertian nor BRDF.

   **Reply**: Thank you for your suggestion. Yes, Eq (1) and Eq (2) are good for specular reflection, and have been used in similar referenced work, so we used them to calculate

surface emissivity in this work. The results derived from this assumption will be investigate more in the later by combining more **auxiliary** observations over the actual surface of test plots. We added more sentences about this comment in line 293-300 as following "**It is noted here that Eq.(1) is** assumed for specular reflection, and was used in previous similar observation study (**Lemmetyinen et al., 2015; Montpetit et al., 2018)**, so we used Eq.(1) and (2) to calculate surface emissivity in this work. The dual-polarized radiometer can provide both vertical and horizonal polarization information, then the idea and uniform **Lambertian surface is too simple and the bidirectional reflectance (BRDF) surface seems more complex, and the specular reflection is a good option. The** results derived from this assumption will be further investigated by combining more **auxiliary** observations in the actual surface of test plots."

8. The brightness temperature change for cement and sand in (a) of Fig. 5 follows the change of the surface temperature. But the brightness temperature for soil and grass between 8 and 12 looks strange. The authors can use the data in Fig. 5 to derive the surface emissivity.
   **Reply**: Thank you for your comment. Yes, the surface emissivity shown in this paper is calculated from the observed Tb used in Fig.5. As to brightness temperature change in Fig.5, it is a monthly averaged value in Oct.2018, so the observed Tb over soil and grass between 8-12hr look less smoothly change than that over cement and sand, and the change pattern within 24hr looks similar to the surface temperature.

9. The y-axis titles in (b) of Fig.(5), (c) of Fig.(6), (b) of Fig.(7), and (c) of Fig. (8) aren't right. The title should be "Brightness temperature difference" or "Emissivity difference".
   **Reply**: Thank you for your suggestion, the titles have been corrected.

**Citation**: https://doi.org/10.5194/amt-2021-165-RC3

---

## Author Response (AR2)

Thank you for addressing the reviewer comments. It is my judgement that they have been satisfied in the revisions, but I noticed a few remaining technical or language errors in the revised manuscript. Please make these corrections to complete the manuscript for publication.

Line 293: "study" should be "studies"
Reply: Thank you for your comments. Yes, we did the correction, please see in the paper "It is noted here that Eq.(1) is assumed for specular reflection, and was used in previous similar observation studies (Lemmetyinen et al., 2015; Montpetit et al."

Line 296: "idea" should be "ideal"
Reply: Yes, we did the correction, please see in the paper" information, then the ideal and uniform Lambertian surface is too simple"

Line 317: "shown" should be "shows"
Reply: Yes, it was corrected as "As Fig.5a shows"

Line 369: "corresponding" should be "corresponds to", and this sentence is missing a period at the end.
Reply: Yes, it was corrected as "Fig. 6 Variations in the observed Tb over different land surfaces(a) and water surface(b) as well as Tb polarized difference(c) with the elevation angle in October 2018. The vertical dotted line corresponds to elevation angle 36°"

Line 387: "is" should be "are"
Reply: Here it was corrected as "Emissivity polarization difference is"

Line 389: "property" should be "properties"
Reply: Yes, it was corrected as "dielectric constant properties."

Line 393: Is "scatters" in reference to volume scattering by vegetation or Lambertian surface scattering? Please be specific.
Reply: Thank you for your comments, it was corrected as "Both Tb and emissivity

polarized difference demonstrated that surface roughness over grass is obviously larger than that over other three land surfaces, especially smooth cement surface, thus generate more volume scattering by vegetation and weakens the polarization difference over grass."

Line 398: This sentence needs a period

Reply: Thank for your reminding. This sentence is corrected as "Fig. 8 Variations in the surface emissivity (a, b) and $\varepsilon$ polarization differences(c) over different surfaces with increasing elevation angle in Oct. 2018."